# Legionellosis-Associated Hospitalization in Spain from 2002 to 2021

**DOI:** 10.3390/microorganisms11071693

**Published:** 2023-06-29

**Authors:** Enrique Gea-Izquierdo, Ruth Gil-Prieto, Valentín Hernández-Barrera, Gil Rodríguez-Caravaca, Ángel Gil-de-Miguel

**Affiliations:** 1Preventive Medicine and Public Health, Rey Juan Carlos University, 28922 Madrid, Spain; 2María Zambrano Program, European Union, Spain; 3Department of Preventive Medicine, Hospital Universitario Fundación Alcorcón, Rey Juan Carlos University, 28922 Madrid, Spain; 4CIBER of Respiratory Diseases (CIBERES), Instituto de Salud Carlos III, 28029 Madrid, Spain

**Keywords:** legionellosis, Legionnaires’ disease, hospitalization, COVID-19, Spain

## Abstract

Legionellosis is a respiratory disease of bacterial and environmental origin that usually presents two distinct clinical entities, “Legionnaires’ disease” (LD) and “Pontiac fever”. LD is an important cause of hospital-acquired pneumonia (HAP). The objective of this study is to describe the epidemiology of legionellosis-associated hospitalization (L-AH) in Spain from 2002 to 2021 and the burden of hospitalization due to legionellosis. Discharge reports from the Minimum Basic Data Set (MBDS) were used to retrospectively analyze hospital discharge data with a diagnosis of legionellosis, based on the ICD-9-CM and ICD-10-CM diagnosis codes, from 2002 to 2021. 21,300 L-AH occurred throughout the year during 2002–2021. The incidence of hospitalization associated per 100,000 inhabitants by month showed a similar trend for the 2002–2011, 2012–2021, and 2002–2021 periods. In Spain, during 2002–2021, the hospitalization rate (HR) in the autonomous communities ranged from 4.57 (2002–2011) to 0.24 (2012–2021) cases per 100,000 inhabitants. The HR of legionellosis in Spain has substantially increased across the 2002–2021 period, and the estimate is consistent with available European data. It is considered that in-depth epidemiological surveillance studies of legionellosis and improvements in the prevention and control of the disease are required in Spain.

## 1. Introduction

Legionellosis varies from a mild flu (Pontiac fever) to Legionnaires’ disease (LD), an atypical, often severe form of pneumonia with a 10% mortality rate [1]. LD is an important cause of community-acquired and hospital-acquired pneumonia (CAP and HAP, respectively). Globally, most cases relate to the causative agent *Legionella pneumophila* (LP), although cases related to other *Legionella* bacteria might not have been identified [2]. More than 90% of infections are caused by LP, which is typically inhaled from contaminated environmental aerosols, and most LD cases are sporadic and unlinked to other infections, with many acquired during travel [1]. LD is an important but relatively uncommon respiratory infection that can cause substantial morbidity and mortality. The morbidity associated with LD, its widespread occurrence, and recent major outbreaks emphasize the need for further research to support early diagnosis [2] and improve prevention and control of the disease.

LD is a notifiable condition in all 30 European Union (EU) and European Economic Area (EEA) countries, where ≈70% of reported cases are CAP, ≈20% are travel-associated, and ≈10% are healthcare associated (HCA) [3]. An increase in the number of reports of legionellosis in the European Union and the European Economic Area has been recorded in recent years. The increase in cases is significant: from 6947 reports in 2015 to 11,298 in 2019 [4]. The ECDC report predicts an increase in the notification rate of legionellosis from 1.4 to 2.2 cases per 100,000 inhabitants in the next four years [5].

HCA is linked with nosocomial outbreaks, underdiagnosis, and a high death rate of ≈30%. During 2006–2017, nearly 25% of identified outbreaks in the United States and several countries in Europe occurred in nosocomial settings. During 2005–2009 in the United Kingdom and 2008–2010 in Spain, ≈3–4% of HCA pneumonia cases were caused by *Legionella* [3].

It is known that LD can cause sporadic and epidemic CAP, as well as HAP, in both healthy and immunocompromised hosts. LD accounts for approximately 2–15% of all cases of CAP requiring hospitalization in North America and Europe [6], and recent studies suggest that rates of legionellosis may be increasing [7]. The estimated cost of LD treatment for hospitalized patients in the USA reaches over USD 433 million (2012) [8].

LD is among the most severe and costly waterborne illnesses in the United States, where it is responsible for an estimated 15% of all deaths related to waterborne infectious diseases and between 3 and 9% of all cases of CAP [9]. The reported mortality rate of LD in the EU/EEA (2015) was 0.8 per million inhabitants, which is consistent with the rates recorded since 2008, which ranged from 0.7 to 0.9 per million inhabitants [10]. LD is a common cause of CAP with ~10% mortality; most patients require hospitalization, with some progressing to acute respiratory failure leading to intensive care unit admission, similar to coronavirus disease 2019 (COVID-19) [2].

As SARS-CoV-2 continues to sweep through the world’s population, healthcare providers should be on heightened alert for another potential cause of pneumonia with similar symptoms: LD. Public health professionals have recognized that due to the similarities in initial disease presentation, clinicians may repeatedly test for COVID-19 before recognizing the need to test for LD (Cassell and Davis, 2021). Because clinical manifestations may be indistinguishable between COVID-19 and LD, targeted microbiologic testing for both *Legionella* and SARS-CoV-2 is essential [11].

In addition, it should be noted that during the COVID-19 pandemic, a multitude of investigations related to biological agents have been generated. However, more studies related to *Legionella* during the pandemic are needed. This study aims to describe the epidemiology of legionellosis-associated hospitalization (L-AH) in Spain from 2002 to 2021. Likewise, to estimate the change in *Legionella*-attributable disease in the Spanish population in relation to the implementation of measures for the control of SARS-CoV-2 and the burden of hospitalization due to legionellosis.

## 2. Materials and Methods

### 2.1. Study Design and Population

We investigated the Spanish notification data on legionellosis from 2002 to 2021 regarding overall time trend, content, and data by a retrospective observational study. At the time of the analysis, these statistics were not yet available for 2022; therefore, we used the statistics from 2021 instead. We explored the relevance of the COVID-19 pandemic to the reported case numbers using an interrupted time series period (1 January 2002 to 31 December 2021). In addition, the following were considered during the period: 2002–2011 and 2012–2021. We studied populations of all ages in Spain by reviewing data from the National Surveillance System for Hospital Data (CMBD), including all hospitalizations for 98% of Spanish hospitals [12,13,14]. Cases were identified through the Spanish public health surveillance system, which includes the entire territory and disaggregates through autonomous communities. The study is exempt from being reviewed by a research ethics committee. The personal information of each subject was delivered to the researchers anonymously, in strict compliance with current Spanish and European legislation. This study complies with the Declaration of Helsinki.

### 2.2. Data Sources

CMBD raw data reports all legionellosis notifications based on the case definition, including only confirmed and probable cases with residency in Spain. The legionellosis notification dataset contained cases notified on any given day. The Spanish Minimum Basic Data Set was implemented in 2016 as a new data model of the Minimum Basic Data Set for Hospital Discharges, extending the registry to other alternative areas of hospitalization (day hospitals, highly complex techniques and procedures offices and emergencies) and to the private sector. We used discharge reports from the CMBD published annually by the Spanish Ministry of Health (SMH) to retrospectively analyze hospital discharge data. In our exploration of the COVID-19 impact, we collected information from CMBD on the pandemic. Data on the number of COVID-19 cases, hospitalizations, and deaths were extracted through SMH requests. Data on the COVID-19 Spanish pandemic contain information from 1 January 2019 to 31 December 2021.

### 2.3. Demographic and Clinical Data

The Spanish CMBD contains routine hospital data, including a diagnosis of legionellosis. All the cases confirmed legionellosis infection, legionellosis testing, or legionellosis-associated respiratory disease. The principal reason for hospitalization was the primary diagnosis code, and admission was considered a single hospital episode. Double-counting admissions that contained several of these diagnoses was avoided. The annual hospitalization rate (HR), average length of hospitalization stay (LOHS), and case-fatality rate (CFR) were calculated using municipal register data. All L-AH in any diagnostic position with the International Classification of Diseases, 9th revision, Clinical Modification ICD-9-CM codes (“482.8” and “482.84”) and ICD-10-CM codes (“A48.1” and “A48.2” from 2016 to 2021) were analyzed. For each case, specific data were gathered on age, sex, average LOHS, HR, and CFR. Groups of age for the study were 0–4, 5–14, 15–24, 25–34, 35–44, 45–54, 55–64, 65–74, 75–84, and >85. First-incidence hospitalizations with a diagnosis of legionellosis were included in the study. In rates by months of age, the number of total newborns and perinatal mortality in Spain were also considered through the Spanish Instituto Nacional de Estadística (INE). For the population denominators by months of age, we assumed a constant birth rate throughout the year to associate the precise hospitalization date for the population at risk in any given month. The CFR was calculated by dividing the number of deaths by the total number of hospitalizations due to legionellosis (%). The average LOHS was calculated [15]. The datasets generated and/or analyzed during the current study are available in the Hospital Discharge Records in the Spanish CMBD repository.

The case notification contains patient demographic information, clinical and diagnostic reports, about the notification process, the severity of the disease, the risk of mortality for the patient, and its impact on the cost of the service. For each record, the following variables were collected: year, autonomous community, Center Type N4, SNS/not SNS, hospital group, cluster group, age, sex, country of birth, country residence, autonomous community residence, province residence, municipality residence, start date contact, date of admission, end date contact, intervention date, contact type, hospital discharge type, origin, contact circumstance, care continuity, ICU admission, ICU days, stay days, principal diagnostic, diagnostics 2–20, POA main diagnosis, POA diagnostics 2–20, procedures 1–20, external procedures 1–6, GRD APR, CDM APR, GRD APR type, APR severity level, APR mortality risk, Spanish weight APR, and APR cost.

### 2.4. Hospitalization Data

The cost of hospitalizations was provided by the Ministry of Health based on the total cost, the number of discharges, and the diagnostic cost group. The last was based on the diagnosis-related groups (DRG) for hospitalized patients depending on discharge ICD classification. DRG calculations were made by 3MTM with Core Grouping System Software (3M Health Information Systems. 3M HIS Murray, UT 84123 USA).

### 2.5. Statistical Analysis

The chi-square test was used to assess differences in proportions, and Student’s *t*-test was used for comparing continuous variables. Poisson regression was used to assess differences in HR during the study period in all the age groups. Joinpoint regression models were used to evaluate trends in hospital admissions for legionellosis over the study period [16]. We calculated the annual percentage change (APC) in HR between the trend change points and estimated the average APC (AAPC) over the 2002–2021 period. When there are no changes in the trend (no joinpoints), APC is constant and equals AAPC. In all tests, the significance level used was *p* < 0.05. For the statistical analysis, we used R (Version 4.0.3. R Core Team, 2020) and Joinpoint Software, Version 4.8.0.1 (National Cancer Institute, National Institutes of Health, Bethesda, MD, USA).

## 3. Results

The dataset of legionellosis hospitalizations for the 2002–2021 period covered a total of 21,300 notified cases in CMBD. There were 10,463 cases in 2002–2011 and 10,837 cases in 2012–2021. In cases comprised of 74.78% (N = 15,928) males and 25.22% (N = 5372) females, the median age was 61 and 68 years old, respectively. The age group of 60–69 years old made up nearly one-quarter of the legionellosis cases (21.2%). The overall mean annual incidence rate for hospitalizations for the whole period (2002–2021) was 2.38 (95%CI = 2.35–2.41), while 2002–2011 and 2012–2021 were 2.41 (95%CI = 2.36–2.46) and 2.36 (95%CI = 2.32–2.4) hospitalizations per 100,000 population, respectively (Table 1). Hospitalization admissions for patients with legionellosis diagnosis significantly decreased after 2012 compared with years 2002–2011. These rates showed a slight linear decrease of 0.41% per year during the study period (*p* < 0.001). The overall period males’ HR were more than double those for females, at 3.62 (95%CI = 3.56–3.68) versus 1.19 (95%CI = 1.16–1.22) per 100,000 population. The HR was highest for the >85 years old, 7.6 (95%CI = 7.24–7.96), with 7.1 (6.54–7.66) and 7.91 (7.44–8.38) for the 2002–2011 and 2012–2021 periods, respectively. The incidence of hospitalization associated with legionellosis per 100,000 inhabitants by month in Spain showed a similar trend for the 2002–2011, 2012–2021, and 2002–2021 periods. In all of them, there was an incidence increase from April to September (Figure 1). In 2002–2021, LOHS significantly increased with age, being 11.95 days in the population from 15–24 years old (SD 4,1) (95%CI = 9.02–14.89) and 65–74 years old (SD 5,3) (95%CI = 11.4–12.5) (*p* < 0.001) (Table 2). Average LOHS were higher in females at 11.1 (95%CI = 10.75–11.45) than in males at 10.65 (95%CI = 10.43–10.86). The median number of hospitalization dates was 10.76 days. In 2002–2021, the median days remained stable, particularly during the peaks of the pandemic (spring and autumn 2020). Average hospital admissions per season in Spain in the 2002–2011, 2012–2021, and 2002–2021 periods are shown in Figure 2. Clearly, it can be seen that in all of them, summer and winter were the seasons with the most hospitalizations.

Over 2002–2021, the autonomous community of Basque Country accounted for 4.24 (95%CI = 4.05–4.43) hospitalizations per 100,000 population, followed by the autonomous community of Navarra at 4.06 (95%CI = 3.71–4.41) and Cantabria at 4.02 (95%CI = 3.65–4.39) (Table 3). In this period, the autonomous community of Navarra reported the highest HR of 4.57 (95%CI = 4.03–5.11) compared to the overall mean. In contrast, fewer HR were reported from the autonomous communities of Melilla 0.28 (95%CI = 0–0.67) and Extremadura 0.67 (95%CI = 0.52–0.82). In the last 10-year period (2012–2021), the autonomous community of Cantabria reported the highest HR of 4.14 (95%CI = 3.62–4.66) compared to the overall mean. Fewer HR were reported from the autonomous communities of Ceuta 0.24 (95%CI = 0–0.57) and Melilla 0.48 (95%CI = 0.01–0.95). Using joinpoint regression analysis, no evidence of statistically significant changes in the mean annual HR due to legionellosis or changes in the trend was found throughout the 2002–2021 period analyzed in: Asturias (APC = −0.34), Balearic Islands (APC = 0.73), Cantabria (APC = 2.12), Castilla Leon (APC = −0.22), Castilla La Mancha (APC = 3.90), Community of Madrid (APC = 0.50), Navarre (APC = −1.30), Basque Country (APC = −0.21), La Rioja (APC = 0.91), Ceuta (APC = −21.10), and Melilla (APC = 30.56). However, joinpoint regression analysis showed a statistically significant increase in the mean annual HR for legionellosis (4.3% per year, *p* < 0.05), with no changes in the trend over the study period in Extremadura. Finally, the model showed a statistically significant increase */decrease ** in the mean annual HR for legionellosis (% per year, *p* < 0.05), with changes in the trend over the study period in Spain (2002–2014, −3.35% **; 2014–2021, 8.74% *), and autonomous communities of Andalusia (2016–2021, 19.1% *), Aragon (2004–2011, 15.1% **; 2011–2021, 8% *), Canary Islands (2004–2014, 12.1% **; 2014–2021, 14.3% *), Catalonia (2002–2013, 4.8% **), Community of Valencia (2005–2013, 7.5% **; 2013–2021, 6.3% *), Galicia (2012–2021, 9.1% *), and Murcia (2016–2021, 26.2% *) (Figure 3). During the 20-year study period, LOHS were higher in Melilla at 19.17 (95%CI = 9.89–11.61) with Canarias at 16.87 (95%CI = 12.94–12.61) in the second position. The same position could be found in LOHS by sex (*p* < 0.001) (Table 4). During this period, the Basque Country showed the highest LOHS in males of 6.69 (95%CI = 6.34–7.04) and Navarra the highest LOHS in females of 2.47 (95%CI = 2.08–2.86).

The overall CFR of the whole period (2002–2021) was 6.45 (95%CI = 6.13–6.78), ranging from 6.24 (95%CI = 5.79–6.71) in 2002–2011 to 6.66 (95%CI = 6.2–713) in 2012–2021. CFR in females was higher than in males, at 7.28 (95%CI = 6.61–7.99) versus 6.17 (95%CI = 5.81–6.55). The CFR was highest for the 0–4 years old, 22.22 (95%CI = 8–44.58), with 10 (95%CI = 1.1–38.13) and 37.5 (95%CI = 11.9–70.52) for the 2002–2011 and 2012–2021 periods, respectively (Table 1). During the 20-year study period, CFR was highest in Ceuta at 14.29 (95%CI = 0–32.62) with Aragon at 10.52 (95%CI = 8.58–12.46) in the second position. During this period, Ceuta showed the highest CFR in males at 18.18 (95%CI = 0–40.97) and Extremadura the highest CFR in females at 14.29 (95%CI = 4.49–24.09) (*p* < 0.001) (Table 3).

The total cost for the entire 20-year period was EUR 141,6 million, with an average hospitalization cost per case of EUR 6510. In 2002–2011 and 2012–2021, the total cost was EUR 68.5 million with an average hospitalization cost per case of EUR 6391 and EUR 73 million with an average hospitalization cost per case of EUR 6626, respectively. During the 20-year period, the autonomous community of the Canary Islands showed the highest average hospitalization cost per case of EUR 8928. In 2002–2011 and 2012–2021, the Canary Islands were EUR 8555 and EUR 9378, respectively.

## 4. Discussion

The genus *Legionella*, which comprises a large group of bacteria, poses a serious threat to human health and life. Since 1977, the genus *Legionella* has been recognized as a frequent cause of CAP and a rare cause of HAP. Over time, the prevalence of LD has risen, which might indicate a greater awareness and reporting of the disease [17], but *Legionella* is an overlooked pathogen in HAP [18]. Even prior to the pandemic, legionellosis was a global health concern with an estimated seroprevalence of 13.7% [19]. In 2011, the LD notification rate in the EU/EEA returned to the levels observed from 2005 to 2009, after the peak observed in 2010, when one case per 100,000 population was reported each year, while in the 2011–2015 period, the EU/EEA age-adjusted notification rate steadily increased to reach 1.30 cases per 100,000 population in 2014–2015, the highest rate ever observed [20]. While in about half of the European countries the upward trend in LD case notification after 2018 continued, the summer peak in 2018 could be observed across all the EU/EEA and has been unmatched in 2019. The fact that the USA also reported a similarly high notification rate in 2018 suggests that larger-scale effects affected the occurrence of LD [21].

In the United States, LD incidence rates increased in 2017–2018, among other years. The rise observed in 2017–2019 was relevant to the population aged >50 years old, and this matches the fact that these age groups comprise an increasing proportion of the EU/EEA population. There is an excess and increasing trend in LD for both sexes, with a higher annual proportion of males consistent throughout the period [22]. Graham et al. determined an increase in HR between 2000–2009 (0.56) and 2010–2020 (2.34) per 100,000 population, elevated after 2015, and higher rates of hospitalization in legionellosis male patients >45 years old, for those of European/other ethnicity, and during spring or summer [23]. In 2017, the European LD notification rates increased from 1.8 per 100,000 to 2.2 per 100,000 population in 2018 and 2019. Since 2011, Italy, France, Germany, and Spain have consistently reported the majority of LD cases annually in the EU/EEA, and an increase in LD cases during 2017–2019 was unusual compared with the previous 5-year period. However, there is no explication of the trend but a significant increase in the change in trend (LD cases in 2017 were the start of an increase in comparison to 2012–2016). In Spain, during 2002–2021, the HR in the autonomous communities ranged from 4.57 (2002–2011) to 0.24 (2012–2021) cases per 100,000 inhabitants. The age- and sex-standardized annual incidence reflected similar trends in autonomous communities, as noted. This indicates that differences between communities are probably not the result of the age and sex distributions of the populations in each one. As in other parts of Europe, between 2016 and 2017, most autonomous communities in Spain experienced an unexplained increase in LD, as they did in 2018 as well. Perhaps the increase in LD observed in 2017 followed a general trend that began in 2011. The age groups 65–74 and 85 years and older showed large increases in LD incidence in 2017 and 2018 compared with 2015 and 2016 rates, but the reasons for this are unclear. An underlying sex distribution among older adults was identified [24].

Our surveillance study suggests that the HR of legionellosis in Spain has substantially increased across the assessed 2002–2021 period, and our estimates are also quite consistent with available European data [25]. To date, this upsurge is a common and not well-understood feature of nearly all high-income countries and could be advocated as an explanation for the extensive heterogeneity among the prevalence data managed. Additionally, we identified geographic trends in HR, with estimates significantly increasing from northwestern regions to northeastern regions and decreasing from the east to central, southern, and southwestern regions. This behavior could be due to different factors, including environmental ones.

The HR of LD decreased during the COVID-19 pandemic in 2020, but several countries still detected and investigated outbreaks. According to ECDC, in 2020, the cause of the higher notification rates observed in Europe in previous years will remain unknown. Contributing factors may include changes in national testing policies and surveillance systems, an aging EU/EEA population, the design and maintenance of infrastructure in building water systems, and changes in climate and weather patterns across Europe and worldwide, which can impact both the ecology of *Legionella* in the environment and the causes of exposure to water aerosols containing the bacteria [26].

As noted, 21,300 L-AH occurred throughout the year during 2002–2021. Nevertheless, a seasonal pattern was observed in the spring (March–May) and summer (June–August). Seasonal trends of L-AH showed a peak in winter in the Canary Islands compared to the rest of the autonomous communities. Figure 2 shows the increasing case numbers from April until September, followed by a small drop from October to March. The annual crude notification rate for legionellosis cases ranged from 0.36/100,000 population in 2002–2011 to 0.1/100,000 population in 2002–2011 and 2012–2021. The highest notification rate was recorded in 2021, with 3.3/100,000 population. There was a strong annual seasonality in the data, peaking around the calendar month of October. The record-high year of 2003 showed a strong peak, which, however, shifted to August instead of October. Since May, the increase in cases has been more pronounced than the increase in the winter months. Comparing the period 2002–2011 with 2012–2021, the number of cases increased most strongly from summer to spring by 91%, compared to an increase of 59% in autumn (September–November) and 8% in winter (December–February).

The steady increase starting in 2014 observed across Spain can be attributed to uncertain changes, changes in test type, or test performance for pneumonia-like diseases, among others [27]. Joinpoint analysis confirmed that a change in trend occurred in Spain between 2002 and 2021, inclusively; a decreasing trend was identified before 2014 (the change point), and an increasing trend was identified after. Although 2014 was indicated as the single optimal change point and hospitalization incidence increased slightly every year after until 2021, the first substantial increase beyond what was likely the baseline range occurred last year. Legionellosis exhibits a summer-through-early-fall seasonality, and this pattern became more pronounced as incidence increased, which could imply that the cyclical factors causing seasonal patterns are becoming more extreme. More legionellosis cases occurred during June–November, especially in the north and northeast, but most pneumonia-associated hospitalizations occurred during July–October [28]. Internationally, a striking epidemiologic feature of LD is its seasonality; more cases are reported during the summer, and there is a trend towards cases occurring in the humid months. Weather—particularly temperature and humidity—drives the summer spike in incidence [29].

Surveillance for LD is crucial to understanding the temporal and geographical trends of the disease [24]. The study demonstrates that national-level data over a 20-year period in Spain can provide important insights into hospitalization trends that could be contrasted with other European countries and long-term periods. However, improved surveillance could be a potential reason for the continued increase in legionellosis incidence across Europe, but not for the changes in seasonal characteristics of the disease.

In the USA, Alarcon et al. showed a notable shift in seasonal peaks in legionellosis from mid-September before 2003 to mid-August after 2003, along with a substantial increase in seasonal amplitude of 97% between 1993 and 2015. The shift in seasonal peaks before and after 2003 was also observed between 1998 and 2006, along with a seasonal amplitude increase of 100% in older adults and 95% in the general population. The shift in disease peak towards mid-August was indicative of changes in the epidemiology of disease transmission, and the incidence of legionellosis has been increasing in the USA since 2003. A seasonal pattern of high incidence in late summer and fall and low incidence in winter was determined with changes in peak timing [27]. In Mediterranean countries, such as Spain, high environmental temperatures in the summer stimulate *Legionella* proliferation. Thus, Spanish guidelines include routine evaluations, but these do not happen in other European countries, such as Germany. This is a very important aspect due to the global increase in temperatures throughout the year [30]. The reported incidence of legionellosis in Spain has been rising since 2014, and the increase appears to be accelerating in recent years. In total, 836 legionellosis cases were reported in 2014. In June, an early peak in legionellosis cases could be seen (1829 cases), followed by the expected seasonal increase in cases by October. The number of legionellosis cases followed the usual seasonality, with more cases occurring in the summer than in the winter. This contrasted with the period of relatively low COVID-19 incidence before the surge of the “second wave”.

Sometimes clinical manifestations may be indistinguishable between COVID-19 and LD. In the COVID-19 pandemic, all pneumonia patients must be admitted via emergency room to isolated wards and tested for COVID-19 repeatedly. As indicated by the American Thoracic Society (ATS) and the Infectious Diseases Society of America (IDSA) guidelines for CAP, only severe cases should be tested with the *Legionella* urinary antigen test or those with epidemiological indications [31]. Despite the fact that the cumulative prevalence of 0.288% in the entirety of SARS-CoV-2 cases may appear quite irrelevant, it would correspond to an unprecedented number of diagnoses for *Legionella* infections in the general population. In some contexts, the estimated burden of *Legionella* infection for the time period from March to December 2020 exceeds several times the official notification rates for 2020. Other times, 2020 notification rates for *Legionella* have slightly but substantially decreased in most European countries, presumptively because of the travel ban implemented during the “first wave” of the pandemic and the subsequently enforced restrictions on international travel. We cannot rule out that the very high prevalence of *Legionella* among SARS-CoV-2 patients may be influenced by the “true” occurrence of this pathogen in the general population and consider *Legionella* infections notoriously underestimated [32].

Despite the fact that productivity losses caused by LD account for just more than half of its lifetime economic burden, almost all of which (95%) are caused by productivity losses from premature deaths (Baker-Goering, 2014), in Spain it represents a health impact to consider and an improvement in disease prevention and epidemiological monitoring. The economic burden of LD more than doubles when lifetime productivity losses are added to medical costs. Estimates have demonstrated the value of investments in preventing LD, such as water management programs and outbreak investigations [33].

Our analyses were subject to certain limitations. First, the characterization of seasonality has been limited to the description of the month with a high incidence of hospitalization. This could be imprecise and reduce the ability to compare seasonal variations that can be identified by the average hospital admissions per season. Second, testing intensity fluctuations could also contribute to the increase in reported legionellosis and autonomous communities’ differences in incidence.

A better understanding of the clinical presentation and prognostic factors for legionellosis may optimize our therapeutic approach and management of the disease and thus reduce mortality and morbidity. However, it should be noted that with the increase in age and highly susceptible population, the observed trends of legionellosis could be alarming in the future. Improved investigations of sporadic cases and their sources [34,35] may lead to novel prevention strategies and the identification of previously unrecognized outbreaks.

## Figures and Tables

**Figure 1 microorganisms-11-01693-f001:**
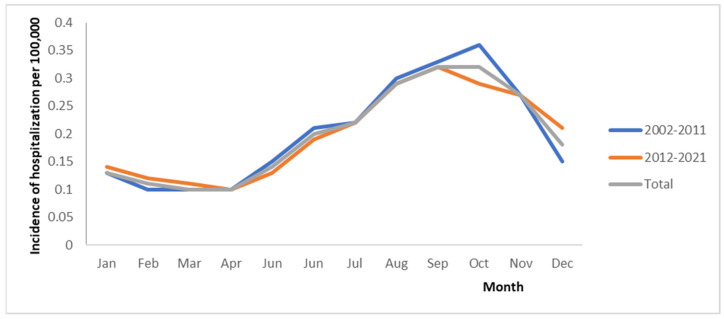
Incidence of hospitalization associated with legionellosis per 100,000 inhabitants by month in Spain.

**Figure 2 microorganisms-11-01693-f002:**
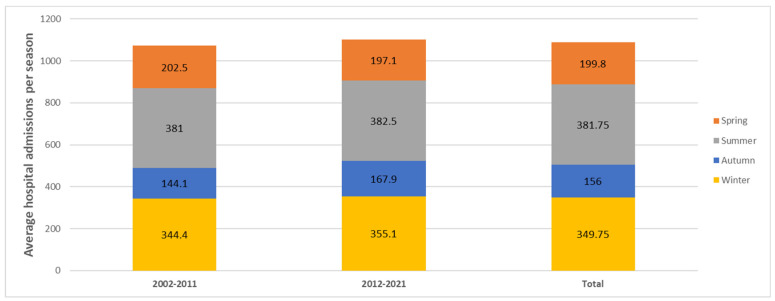
Distribution of the average hospital admissions per season of legionellosis in Spain, from 2002–2011, 2012–2021, and 2002–2021.

**Figure 3 microorganisms-11-01693-f003:**
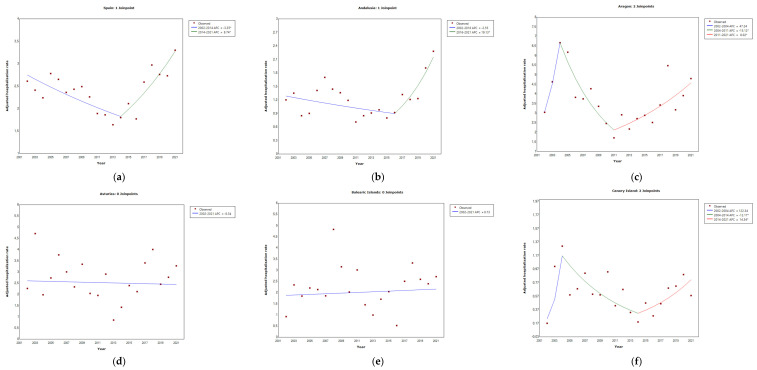
Joinpoint regression analysis of the adjusted hospitalization rates for legionellosis: Spain (**a**), Andalusia (**b**), Aragon (**c**), Asturias (**d**), Balearic Islands (**e**), Canary Islands (**f**), Cantabria (**g**), Castilla Leon (**h**), Castilla La Mancha (**i**), Catalonia (**j**), Community of Valencia (**k**), Extremadura (**l**), Galicia (**m**), Community of Madrid (**n**), Murcia (**o**), Navarre (**p**), Basque Country (**q**), La Rioja (**r**), Ceuta (**s**), and Melilla (**t**) by year, from 2002–2021. Asterisk indicates that the annual percentage change (APC) is significantly different from zero at the alpha = 0.05 level.

**Table 1 microorganisms-11-01693-t001:** Hospitalization rate (HR) and case fatality rate (CFR). Legionellosis by year of age and sex in Spain (2002–2021).

Age (Years)	HR in Males	HR in Females	*p*-Value	CFR in Males	CFR in Females	*p*-Value
0–4	0.03 (0.01–0.05)	0.05 (0.02–0.08)	0.638	37.5 (11.9–70.52)	10 (1.1–38.13)	0.188
5–14	0.02 (0.01–0.03)	0.02 (0.01–0.03)	0.493	0 (0–0)	0 (0–0)	-
15–24	0.13 (0.1–0.16)	0.08 (0.05–0.11)	0.040	2.99 (0.63–9.23)	8.11 (2.34–20.08)	0.261
25–34 *,+,¥	0.72 (0.66–0.78)	0.17 (0.14–0.2)	<0.001	1.66 (0.79–3.11)	2.65 (0.75–6.91)	0.484
35–44 *,+,¥,†,§	2.28 (2.17–2.39)	0.46 (0.41–0.51)	<0.001	2.91 (2.19–3.78)	1.8 (0.76–3.68)	0.262
45–54 *,+,¥	4.98 (4.81–5.15)	1.15 (1.07–1.23)	<0.001	2.4 (1.92–2.96)	4.07 (2.84–5.65)	0.011
55–64 *,+,¥	7.46 (7.22–7.7)	1.97 (1.85–2.09)	<0.001	4.33 (3.72–5.01)	4.25 (3.16–5.59)	0.912
65–74 *,+,¥	8.4 (8.11–8.69)	2.64 (2.49–2.79)	<0.001	6.73 (5.9–7.63)	5.95 (4.7–7.42)	0.355
75–84 *,+,¥	11.09 (10.67–11.51)	3.69 (3.49–3.89)	<0.001	11.85 (10.66–13.12)	10.2 (8.61–11.97)	0.131
>85 *,+,¥	12.9 (12.08–13.72)	5.04 (4.68–5.4)	<0.001	17.23 (14.93–19.74)	14.95 (12.57–17.6)	0.209
Total *,+,¥	3.62 (3.56–3.68)	1.19 (1.16–1.22)	<0.001	6.17 (5.81–6.55)	7.28 (6.61–7.99)	0.010

HR (Hospitalization rate): * Significative difference by year in male (*p* < 0.001). + Significative difference by year in female (*p* < 0.001). ¥ Significative difference by year (*p* < 0.001). CFR (case fatality rate): † Significative difference by year in male (*p* < 0.001). § Significative difference by year (*p* < 0.001).

**Table 2 microorganisms-11-01693-t002:** Legionellosis by length of hospitalization stay (LOHS) by age in Spain (2002–2021).

Age (Years)	LOHS in Males	LOHS in Females	*p*-Value
0–4 years	9.88 (2.71–17.04)	13.3 (6.56–20.04)	0.437
5–14 years	9.09 (4.52–13.66)	8.38 (5.22–11.53)	0.792
15–24 years	10.82 (6.74–14.9)	14 (10.16–17.84)	0.306
25–34 years	8.3 (7.47–9.13)	10.29 (7.58–13.01)	0.07
35–44 years *,¥	9.47 (8.93–10.02)	10.91 (9.18–12.64)	0.052
45–54 years *,¥	10.41 (9.9–10.92)	11.45 (10.43–12.47)	0.08
55–64 years *,¥	10.39 (10.01–10.76)	10.55 (9.71–11.39)	0.703
65–74 years *,+,¥	11.87 (11.2–12.54)	12.18 (11.27–13.1)	0.625
75–84 years *,+,¥	11.36 (10.93–11.79)	10.97 (10.36–11.59)	0.318
85 o mas *,¥	9.72 (9.21–10.22)	10.17 (9.65–10.69)	0.222
Total *,+,¥	10.65 (10.43–10.86)	11.1 (10.75–11.45)	0.035

* Significative difference by year in male (*p* < 0.001). + Significative difference by year in female (*p* < 0.001). ¥ Significative difference by year (*p* < 0.001).

**Table 3 microorganisms-11-01693-t003:** Hospitalization rate (HR) and case fatality rate (CFR). Legionellosis by autonomous community and sex in Spain (2002–2021).

Age (Years)	HR in Males	HR in Females	*p*-Value	CFR in Males	CFR in Females	*p*-Value
Andalusia +,†,§	1.88 (1.79–1.97)	0.59 (0.54–0.64)	<0.001	9.97 (8.47–11.47)	11.16 (8.35–13.97)	0.452
Aragon *,¥	5.29 (4.89–5.69)	2.07 (1.82–2.32)	<0.001	9.45 (7.26–11.64)	13.24 (9.21–17.27)	0.086
Asturias *	4.34 (3.93–4.75)	1.15 (0.95–1.35)	<0.001	5.94 (3.73–8.15)	7.14 (2.64–11.64)	0.621
Balearic Islands *,+	3.09 (2.76–3.42)	1.39 (1.17–1.61)	<0.001	8.11 (5.18–11.04)	8.67 (4.17–13.17)	0.837
Canary Islands *	0.91 (0.78–1.04)	0.34 (0.26–0.42)	<0.001	8.11 (4.18–12.04)	7.25 (1.13–13.37)	0.821
Cantabria	6.13 (5.48–6.78)	2 (1.64–2.36)	<0.001	4.35 (2.2–6.5)	3.39 (0.12–6.66)	0.652
Castilla Leon ‡,§	2.33 (2.14–2.52)	0.73 (0.62–0.84)	<0.001	8.07 (5.83–10.31)	9.89 (5.55–14.23)	0.444
Castilla La Mancha *,+,¥	2.68 (2.45–2.91)	0.88 (0.75–1.01)	<0.001	7.41 (5.2–9.62)	5.14 (1.87–8.41)	0.305
Catalonia +,¥,§	6.03 (5.85–6.21)	2.02 (1.92–2.12)	<0.001	4.78 (4.14–5.42)	6.75 (5.48–8.02)	0.003
Community of Valencia *,¥	4.83 (4.63–5.03)	1.73 (1.61–1.85)	<0.001	6.16 (5.18–7.14)	8.61 (6.72–10.5)	0.016
Extremadura *,+,¥	1.37 (1.15–1.59)	0.45 (0.32–0.58)	<0.001	8.84 (4.25–13.43)	14.29 (4.49–24.09)	0.280
Galicia +,¥,†,§	3.54 (3.31–3.77)	0.67 (0.57–0.77)	<0.001	4.82 (3.45–6.19)	4.21 (1.35–7.07)	0.717
Community of Madrid +	2.02 (1.91–2.13)	0.67 (0.61–0.73)	<0.001	6.29 (4.93–7.65)	6.44 (4.13–8.75)	0.914
Murcia +,¥	2.91 (2.63–3.19)	0.99 (0.83–1.15)	<0.001	6.24 (3.92–8.56)	7.14 (2.87–11.41)	0.706
Navarre +,¥,§	5.66 (5.07–6.25)	2.47 (2.08–2.86)	<0.001	5.41 (3.04–7.78)	3.87 (0.83–6.91)	0.463
Basque Country *,+	6.69 (6.34–7.04)	1.92 (1.74–2.1)	<0.001	4.05 (3.02–5.08)	3.06 (1.42–4.7)	0.350
La Rioja	5.04 (4.25–5.83)	1.64 (1.19–2.09)	<0.001	4.49 (1.24–7.74)	3.92 (0–9.25)	0.854
Ceuta *,¥,†,§	1.36 (0.56–2.16)	0.38 (0–0.81)	<0.001	18.18 (0–40.97)	0 (0–0)	-
Melilla	0.38 (0–0.81)	0.39 (0–0.83)	0.999	0 (0–0)	0 (0–0)	-
Total *,+,¥	3.62 (3.56–3.68)	1.19 (1.16–1.22)	<0.001	6.17 (5.8–6.54)	7.28 (6.59–7.97)	0.023

HR (Hospitalization rate): * Significative difference by year in male (*p* < 0.001). + significative difference by year in female (*p* < 0.001). ¥ Significative difference by year (*p* < 0.001). CFR (case fatality rate): † Significative difference by year in male (*p* < 0.001). ‡ Significative difference by year in female (*p* < 0.001). § Significative difference by year (*p* < 0.001).

**Table 4 microorganisms-11-01693-t004:** Legionellosis by length of hospitalization stay (LOHS) by autonomous community in Spain (2002–2021).

Age (Years)	LOHS in Males	LOHS in Females	*p*-Value
Andalusia *,¥	12.4 (11.73–13.06)	12.71 (11.48–13.93)	0.758
Aragon *,+,¥	13.68 (12.59–14.77)	13.28 (11.92–14.65)	0.752
Asturias *,+,¥	9.96 (9.16–10.76)	10.44 (8.73–12.16)	0.04
Balearic Islands	12.23 (10.59–13.87)	11.25 (9.32–13.18)	0.475
Canary Islands	17.03 (11.94–22.12)	16.43 (11.52–21.35)	0.665
Cantabria +,¥	10.88 (9.51–12.25)	12.72 (9.78–15.66)	0.291
Castilla Leon	12.98 (11.92–14.04)	12.51 (10.11–14.91)	0.461
Castilla La Mancha *,+,¥	12.07 (10.76–13.37)	10.74 (9.36–12.11)	0.582
Catalonia *,+,¥	9.19 (8.88–9.5)	10.24 (9.53–10.94)	0.016
Community of Valencia	9.67 (9.22–10.12)	9.88 (9.13–10.64)	0.004
Extremadura	11.7 (10.19–13.21)	16 (10.38–21.62)	0.36
Galicia *,¥	11.28 (10.52–12.03)	12.21 (10.44–13.97)	0.597
Community of Madrid	13.17 (11.61–14.73)	11.66 (10.4–12.93)	0.054
Murcia	9.51 (8.39–10.63)	10.61 (8.95–12.28)	0.781
Navarre	9.74 (8.81–10.67)	9.28 (8.21–10.35)	0.15
Basque Country +,¥	8.83 (8.31–9.36)	9.96 (8.74–11.17)	0.767
La Rioja	8.63 (7.15–10.12)	13.61 (5.79–21.43)	0.798
Ceuta+	8.91 (5.32–12.49)	14.67 (−18.41–47.75)	0.696
Melilla	17.67 (3.54–31.79)	20.67 (−10.39–51.73)	0.259

* Significative difference by year in male (*p* < 0.001). + Significative difference by year in female (*p* < 0.001); ¥ Significative difference by year (*p* < 0.001).

## Data Availability

All of the data generated or analyzed during this study are included in this published article and are available upon reasonable request to the corresponding author.

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
