# Peer review of "Legionellosis-Associated Hospitalization in Spain from 2002 to 2021"

_microorganisms, 2023, doi:10.3390/microorganisms11071693_

Round 1

Reviewer 1 Report

The submission point one of the most important microrganism that is associated to hospitalization worldwide. The only remark to this study is the quality of Figure 3. This point must be improve a lot.

Reviewer 2 Report

Estimated Authors,

I've read with great interest your report on the occurrence of hospitalizations following LD in Spain. The report is well documented, and is highly consistent with other European reports, as accurately discussed in the later sections of the paper.

Data Analysis is not particularly innovative, but it follows a well-standardized approach and is, therefore, quite appropriate.

In fact, my only concerns are focused on the tables and figures, as some improvements are not only recommended, but highly in need.

To begin with:

Table 1, 2, 3: 

- please include (also in the captions of the tables) the meaning of all acronyms at their first appearence;

- the notation for the significance is quite intrusive: please evaluate whether it is in need for the appropriate understanding of the text;

- please also note that p value cannot be equals to 0.000; please change the notation to p < 0.001;

- please explain whether the p values are for b vs. c column and e vs. f column in table 1 and 3, and b vs. c in table 2; if other comparisons are reported, it must be more precisely discussed in the caption of the table.

Figure 1 & 2 could be improved by increasing the size of the fonts

Figure 3 must be reworked by increasing the size of the sub-pictures. Otherwise are not correctly readable. 

Table 4: explain the caption the meaning of LOHS

After these amendments, I'm confident that the paper could be reasonably accepted for publication.

Nothing to be particularly stressed. All the minor typos across the main text (very few, indeed) could be addressed by post-acceptance English editing.

Reviewer 3 Report

Legionellosis-associated hospitalization in Spain from 2002 to 2021” analyzed the incidence of hospitalization associated with legionellosis and the distribution of the average of hospital admissions per season. The paper is well organized and discussed, data considered into the evaluation are significative and statistically evaluated. Figure 3 should be improved; it is not readable in its present form. There are few language mistakes in paper, I would recommend a native speaker for proofreading of the paper

Mother tongue revision

Reviewer 4 Report

The authors describe the epidemiology of legionellosis-associated hospitalization in Spain from 2002 to 2021 and the burden of hospitalization due to legionellosis. The authors report that hospitalization incidence increase in Spain during the summer with a trend towards cases occurring in the humid months, as internationally demonstrated; and in particular as temperature and humidity drive the summer spike in incidence. Some information is necessary to complete the feature of data.

Have the authors analyzed how many cases are hospital-acquired? if there are cases travel-associated and clusters among the cases?   

Another point to underline is the importance of monitoring that allows for the analysis of the conditions of the hospital’s water system as indicated in the following paper to include in the references.

Arrigo, I.; Galia, E.; Fasciana, T.; Diquattro, O.; Tricoli, M.R.; Serra, N.; Palermo, M.; Giammanco, A. Four-Year Environmental Surveillance Program of Legionella spp. in One of Palermo’s Largest Hospitals. Microorganisms 2022, 10,764. https://doi.org/ 10.3390/microorganisms10040764

As well as, it is important to emphasize the presence of Lp in prisons, where the Water systems offer the best growth conditions for Lp and support its spread by producing aerosols,  including in the text the following manuscript. 

Cluster of Legionnaires’ Disease in an Italian Prison. Teresa Fasciana Chiara Mascarella, Salvatore Antonino Distefano, Cinzia Calà , Giuseppina Capra, Angela Rampulla, Paola Di Carlo, Mario Palermo and Anna Giammanco. Int. J. Environ. Res. Public Health 2019, 16, 2062; doi:10.3390/ijerph16112062
